# COVID-19 and Male Infertility: Is There a Role for Antioxidants?

**DOI:** 10.3390/antiox12081483

**Published:** 2023-07-25

**Authors:** Thomas Kalfas, Aris Kaltsas, Evangelos N. Symeonidis, Asterios Symeonidis, Athanasios Zikopoulos, Efthalia Moustakli, Chara Tsiampali, Georgios Tsampoukas, Natalia Palapela, Athanasios Zachariou, Nikolaos Sofikitis, Fotios Dimitriadis

**Affiliations:** 1Independent Researcher, 43100 Karditsa, Greece; thomaskalfas@hotmail.com; 2Department of Urology, Faculty of Medicine, School of Health Sciences, University of Ioannina, 45110 Ioannina, Greece; a.kaltsas@uoi.gr (A.K.); kzikop22@gmail.com (A.Z.); ef.moustakli@uoi.gr (E.M.); zahariou@otenet.gr (A.Z.); nsofikit@uoi.gr (N.S.); 3Department of Urology, Faculty of Medicine, School of Health Sciences, Aristotle University of Thessaloniki, 54124 Thessaloniki, Greece; evansimeonidis@gmail.com (E.N.S.); symeaste@gmail.com (A.S.); 4Independent Researcher, 55131 Thessaloniki, Greece; x.tsiampali@gmail.com; 5Department of Urology, Oxford University Hospital NHS Trust, Oxford OX3 7LE, UK; tsampoukasg@gmail.com; 6Medical Faculty, Medical University of Sofia, 1431 Sofia, Bulgaria; nataliapalapela21@gmail.com

**Keywords:** COVID-19, SARS-CoV-2, male infertility, oxidative stress, reactive oxygen species, antioxidants

## Abstract

Coronavirus disease 19 (COVID-19), caused by severe acute respiratory syndrome coronavirus 2 (SARS-CoV-2), jeopardizes male fertility because of the vulnerability of the male reproductive system, especially the testes. This study evaluates the effects of the virus on testicular function and examines the potential role of antioxidants in mitigating the damage caused by oxidative stress (OS). A comprehensive PubMed search examined exocrine and endocrine testicular function alteration, the interplay between OS and COVID-19-induced defects, and the potential benefit of antioxidants. Although the virus is rarely directly detectable in sperm and testicular tissue, semen quality and hormonal balance are affected in patients, with some changes persisting throughout a spermatogenesis cycle. Testicular pathology in deceased patients shows defects in spermatogenesis, vascular changes, and inflammation. Acute primary hypogonadism is observed mainly in severely infected cases. Elevated OS and sperm DNA fragmentation markers suggest redox imbalance as a possible mechanism behind the fertility changes. COVID-19 vaccines appear to be safe for male fertility, but the efficacy of antioxidants to improve sperm quality after infection remains unproven due to limited research. Given the limited and inconclusive evidence, careful evaluation of men recovering from COVID-19 seeking fertility improvement is strongly recommended.

## 1. Introduction

Coronavirus disease-2019 (COVID-19) is caused by severe acute respiratory syndrome coronavirus-2 (SARS-CoV-2), first identified in December 2019 in Wuhan, China [1]. Due to its extreme contagiousness, the World Health Organization (WHO) declared a pandemic on 11 March 2020 [2]. Besides the well-known respiratory detrimental effects, recent evidence links COVID-19 with higher incidence rates of preeclampsia, while some people may develop neurological disorders after contracting the disease [3,4,5]. Men are at higher risk of contracting COVID-19 than women [6]. Therefore, it is critical to investigate how COVID-19 may affect male health and fertility. Several studies have detected an alteration in the hormonal milieu and a decrease in sperm parameters of infected men, indicating testicular damage and defect in the testis’ endocrine and exocrine function [7]. This alteration of testicular function may be the direct effect of the SARS-CoV-2 on the testis, or the indirect impact of the inflammation, the cytokine storm, and the oxidative stress induced by the virus [8]. Since it is now established that the virus can cross the blood–testicular barrier (ΒΤΒ) due to the infection-induced inflammatory response [9,10,11], there is a possibility of viral infection of the testes. In addition, viruses may survive longer in the testis than in other tissues due to immune privilege and regulatory T cells in this area [12]. In a recent study, SARS-CoV-2 was detected in the semen of six individuals, including two who had recovered from COVID-19, according to Li et al. This finding has reignited the discussion about COVID-19 infection of the male reproductive system, virus release in sperm, and possible reproductive therapy for affected individuals [10]. Testicular congestion, interstitial edema, erythrocyte exudation, and infiltration by macrophages and T lymphocytes (CD3+) were detected in the testes of deceased COVID-19 patients.

Increased inflammation was noted in both the testes and epididymis. According to the same studies, men who died of COVID-19 had increased angiotensin-converting enzyme 2 (ACE2) expression in their Leydig cells [13]. Seminiferous tubules from deceased COVID-19 patients show an increase in apoptotic cells, suggesting that spermatogenesis was disrupted by SARS-CoV-2. Viral infection is also known to affect seminal fluid quality and endocrine dyshomeostasis. Ma et al. showed that the ratio of testosterone (T) to luteinizing hormone (LH) was lower, and circulating levels of LH were higher in COVID-19 patients than in the general population [14]. This suggests that the patients had abnormal Leydig cell activity. Koç and Keserolu [15] reported that testosterone levels are lower after infection, a finding that Cinislioglu et al. [16] related to disease severity. The results regarding sperm quality are somewhat contradictory. Sperm quantity, total and progressive sperm motility, and sperm morphology are significantly affected by COVID-19 infection [15,17,18]. It is important to note that impairment of antioxidant activity and induction of oxidative stress (OS) may be responsible for the overall mode of action of respiratory viral infections (a cascade of systemic pathophysiological events, including inflammation, cytokine generation, host cell death, and more) [19]. The positive results of antioxidant therapies warrant further exploration of the possibility of antioxidant treatment of COVID-19 to improve not only COVID-19 but also the disease-related disorders of male reproduction. This review aims to identify the effect of COVID-19 on testicular endocrine and exocrine function, study the possible interplay of COVID-19-induced oxidative stress with male fertility, and explore any potential role of antioxidants in the management of COVID-19.

## 2. Search Strategy

A detailed, nonsystematic electronic search was performed to find evidence of the effects of COVID-19 on male fertility and to investigate the possible role of antioxidants in counteracting COVID-19 oxidative-stress-induced damage in male reproduction. The search was performed in the Medline database using the PubMed search engine. The search terms used were (COVID-19 OR SARS-CoV-2) AND (semen OR sperm OR semen parameters OR hormones OR sex hormones OR testosterone OR gonadotropins OR testis OR testis OR autopsies OR oxidative stress OR reactive oxygen species OR antioxidants OR vaccines) AND (male fertility OR male infertility OR male reproductive system OR hypogonadism). The first search from inception to 24 May 2023, found 443 articles. Based on the title and abstract, 289 articles were deemed appropriate for our search topic. Of these, 98 articles were included in the final text. The references of the eligible articles were manually searched for additional articles relevant to this work’s overall goal. A total of 152 articles were considered to prepare this paper (Figure 1).

Only articles written in English were considered; animal studies were excluded. There were no other restrictions on the types of articles used. Reviews, original research studies, metanalyses, letters to the editor, correspondence, editorials, communications, commentaries, opinions, viewpoints, and perspectives were examined for relevance. A summary of the studies by type can be found in Table 1.

## 3. The Impact of COVID-19 on Male Fertility Explained: Facts, Evidence, and Proof

### 3.1. The Role of ACE-2, TMPRSS-2, and Androgen Receptor

Recently, angiotensin-converting enzyme 2 (ACE-2) was discovered to play a critical role in the respiratory symptoms and pulmonary lesions of SARS-CoV-2 infection by serving as a cell receptor that facilitates SARS-CoV-2 entry into host cells [44,45]. The spike surface glycoprotein (S-protein) of SARS-Cov-2 has a high affinity for ACE-2, a transmembrane zinc metallopeptidase protein bound to the membrane of various tissue cells. It uses it as a receptor for cell entry [46]. The S-protein of the virus binds to ACE-2 and mainly uses transmembrane serine protease-2 (TMPRSS-2), a cellular membrane protease, as a mediator to facilitate fusion with the host cell [20,46]. Due to the high affinity of the S-protein for ACE-2, SARS-CoV-2 can potentially infect any cell expressing ACE-2 protein [20]. ACE-2 expression of SARS-CoV-2 has been detected in many organs, including male gonads [47]. This may also be due to ACE-2, which is more abundant in the testes than in the ovaries. Males have been shown to have a higher risk of developing SARS-CoV-2 compared to females [125]. Activation of the androgen receptor (AR) stimulates the production of TMPRSS-2, a viral spike protein-S1 stimulator, an essential protein for host cell infection, through interaction with ACE-2 [48,142]. Expression of the protease TMPRSS-2 is concentrated in spermatogonia and spermatids in the testis [47]. Recent findings in the prostate cancer study suggest that AR may increase the expression of the TMPRSS-2 gene [49]. Because ACE-2 binds strongly to the outer region of the S1 protein, cells expressing this enzyme are susceptible to infection by SARS-CoV-2 [50]. It can trigger a cascade of viral responses leading to inflammation (e.g., viral orchitis) and, eventually, testicular dysfunction [51]. Examination of the testes of deceased COVID-19 patients revealed significant damage to the testicular ducts, a reduction in Leydig cells, and moderate lymphocytic inflammation. However, analysis at reverse transcription polymerase chain reaction (RT-PCR) revealed the absence of the SARS-CoV-2 virus in the testes of 90% of patients. Electron microscopic examination also failed to detect SARS-CoV-2 [52]. These results suggest that SARS-CoV-2 may play a role, albeit indirectly, in the observed changes in the patient’s testes. Specifically, the inflammatory storm typical of COVID-19 individuals could be responsible for the observed testicular damage. A testicular specimen from a man who tested positive for SARS-CoV-2 showed signs of peritubular fibrosis, which may help explain these findings [143]. In addition, SARS-CoV-2 infection may lead to abnormalities in male reproduction through a pathway triggered by ACE-2, and men with reproductive disorders may be at higher risk of becoming infected with SARS-CoV-2. According to a recent study, testicular ACE-2 mRNA expression is higher in men with infertility [144]. In addition, the same study found that the level of ACE-2 expression in germ cells, Sertoli cells, and Leydig cells correlated with patient age. Patients aged 20–30 years had the highest ACE-2 expression, while patients aged 60 years and older had the lowest [144]. These studies indicate that young men with SARS-CoV-2 infection are at increased risk for testicular damage. The severity and mortality rates of COVID-19 progression varied by sex [53], which may be related to the influence of sex hormones on the body’s response to SARS-CoV-2 infection. Recent clinical studies suggest that a hyperandrogenic phenotype may be associated with increased viral load, greater viral spread, and more severe pulmonary involvement [142], and COVID-19 disease progression has been associated with androgen sensitivity in several studies [146]. The hyperandrogenic phenotype is distinguished by heightened concentrations of androgens, specifically testosterone, which have the potential to enhance the expression of ACE-2 and TMPRSS2. Consequently, this facilitates the infiltration of SARS-CoV-2 into the cellular structures of the host. Simultaneously, the existence of these androgens possesses the capability to diminish the production of specific pro-inflammatory cytokines, thus potentially intensifying the gravity of the infection and its influence on the pulmonary system [142]. T levels naturally decrease with age, which may explain why older men have higher mortality rates [149]. The presence of ACE-2 on the surface of Leydig cells is not only responsible for the testicular damage caused by infection but also has critical hormonal effects: loss of Leydig cells leads to a decrease in T levels and an increase in LH levels, resulting in the occurrence of hypergonadotropic hypogonadism [21], which is associated with a worse prognosis in COVID-19 individuals [54]. Since no alternative human TMPRSS-2 promoter has been discovered to date, it is thought that the activity of AR is required for transcription. Individual differences in androgen sensitivity may be one reason why certain ethnic groups have disproportionately higher rates of disease and mortality in men [146].

### 3.2. SARS-CoV-2 in Human Sperm Samples

The presence of SARS-CoV-2 in the semen of infected patients is controversial. Evidence suggests that the virus can impair the BTB due to an infection-induced inflammatory response, pro-inflammatory cytokine overproduction, and oxidative stress [131]. Although there are no confirmed cases of sexual transmission of the virus, there is evidence of SARS-CoV-2 crossing the BTB and shedding into the semen.

A cohort study by Li et al. showed that SARS-CoV-2 could be present in the semen samples of COVID-19 patients. A total of 38 patients were enrolled in the study; 23 were in the recovery stage, and 15 were in the acute phase. Semen testing found four positive results amongst the acute stage patients (6 to 10 days after onset of symptoms) and two positive results amongst the recovery stage group patients (12 to 16 days after onset of symptoms). One of the limitations is that the study methods were not written in detail [10]. Similarly, a cross-sectional study by Machado et al. included 15 COVID-19-positive men with no or mild symptoms. All samples were collected within two weeks from the onset of the symptoms. The SARS-CoV-2 RNA was detected in the semen sample of one out of 15 patients by RT-PCR [55].

Similarly, a study by Delaroche et al. included 32 SARS-CoV-2-positive patients in the acute phase of the disease (0 to 8 days after onset of symptoms). Twenty-seven patients had moderate symptoms, and five patients were asymptomatic. Only one sample tested positive in the semen sample and seminal plasma. The virus was not detected in the spermatozoa pellet. A slightly higher concentration of bacterial DNA was observed in the SARS-CoV-2-positive specimen, suggesting possible manual or droplet contamination [56]. Another study by Saylam et al. included 30 COVID-19 patients in the acute stage of the disease. Semen urine samples were taken from patients one day after a positive nasal pharyngeal swab for SARS-Co-2. The SARS-CoV-2 was detected in four semen samples and seven urine samples using RT-PCR. The virus was not detected in the semen or urine samples after the patients’ recovery. Patients with virus-positive semen had significantly higher white blood cells (WBC), neutrophils, C-reactive protein (CRP), ferritin, alanine aminotransferase (ALT), lactate dehydrogenase (LDH), and prolactin (*p* < 0.05). Patients with SARS-CoV-2 in semen samples were in the acute stage of the disease, had severe pneumonia, and had a much higher virus load than the other patients, as reported by the authors [57].

In contrast, Kayaslaan et al. reported that no SARS-CoV-2 was found in the semen of 16 men in the acute stage of infection while hospitalized using RT-PCR. The median time between obtaining a semen sample and to positive nasal-pharyngeal swab was one day (0–7). Of the included patients, 5 had moderate disease, and 11 had mild disease (pneumonia) at the time of disease confirmation [58]. Similarly, a study by Rawlings et al. did not detect SARS-CoV-2 RNA in semen samples of six outpatients at the acute or late phase of infection (6 to 17 days after the onset of symptoms). However, the virus was still detected in the saliva samples of all patients [59]. A pilot study by Burke et al. enrolled 18 men with a recent diagnosis of COVID-19 or in the recovery phase. The median time between SARS-CoV-2 detection to semen sample collection was six days (1 to 28). One man was classified as asymptomatic, two as mildly diseased, and fifteen as moderately diseased. The SARS-CoV-2 RNA was not detected in any sample, regardless of the symptoms at the time of semen sample collection [60].

Similarly, a study by Guo et al. examined semen samples from 23 patients in the acute and recovery stage of the disease. A total of 18 patients were diagnosed with a mild type of disease and 5 with moderate disease. All semen samples were tested negative by RT-PCR, while SARS-CoV-2 RNA was still detected in sputum and fecal specimens in 12 patients at the time of semen delivery [61]. Another pilot cohort study by Holtman et al. did not detect the virus in the semen samples of 18 men in the recovery stage and 2 men in the acute phase of infection using the RT-PCR [62]. Additionally, a cohort study by Ruan et al. failed to detect SARS-CoV-2 nucleic acid in the urine, semen, and expressed prostatic secretions (EPS) of 74 men who recovered from COVID-19. The median interval between the last positive nasal-pharyngeal test and sample collection was 80 days. Of all the patients, 11 were classified as mild type disease, 31 were classified as moderate type disease, and 32 had severe pneumonia [63].

Similarly, a cross-sectional study by Pan et al. recruited 34 men recovering from COVID-19. The median time from semen sample collection to COVID-19 diagnosis was 31 days (29 to 36). The patients demonstrated mild to moderate symptoms during the disease; six had scrotal discomfort indicative of orchitis. However, SARS-CoV-2 nucleic acid was not detected in any specimen by RT-PCR [64]. In the same line, many studies that assessed semen for the presence of SARS-CoV-2 during the recovery stage of COVID-19 in non-severe diseased men failed to detect the virus RNA using RT-PCR [14,65,66,67,140].

There are sporadic reports about the virus’s presence in semen though most studies have failed to detect the virus in semen. The diversity between the results reported may be due to differences in disease stage and severity during the semen collection. Viremia is said to be related to severe disease [132]. Thus, the virus may be detected in the acute stage of the disease and in the semen of patients with severe clinical conditions [58]. Therefore, the possibility of sample contamination cannot be ruled out. Nevertheless, there is no evidence of the persistence of the virus in ejaculation at the recovery stage, one month after COVID-19 diagnosis; thus, it is considered very unlikely [61,64]. In conclusion, it is evident from the wide range of detection results that the presence of SARS-CoV-2 in semen samples is an area warranting further investigation (Table 2).

### 3.3. Implications on the Cellular Level

Male infertility and damage to the testes from SARS-CoV-2 have also been reported. Pathologic findings from testicular autopsies of COVID-19 patients have provided much valuable information in recent years about the possible presence of the virus in the testes and the effects at the cellular level that may affect disease progression and fertility in male patients. Tissues from the testes and epididymis of SARS-CoV-2-infected individuals who underwent autopsy showed interstitial edema, congestion, and exudation of red blood cells. More significant numbers of apoptotic cells were also found in testicular tubules, and the researchers noted an increase in the proportion of CD3+ and CD68+ leukocytes in testicular interstitial cells. In addition, increased IL-6, TNF-α, and monocyte chemoattractant protein-1 (MCP-1) were associated with lower sperm counts [13]. Recent studies of ten deceased patients with severe acute respiratory syndrome caused by SARS-CoV-2 showed morphologic changes in the testes and epididymis; changes in spermatocytes, spermatids, and Sertoli cells; and increased OS [133]. These two studies contribute to the growing body of data showing that severe COVID-19 disease can cause SARS-CoV-2 to alter sperm characteristics and exacerbate testicular inflammation [13,133]. On the other hand, the morphologic abnormalities observed in these individuals may be due to the OS and/or the presence of concomitant diseases that occur in many patients.

In contrast to the above studies, a study by Masterson included autopsies of testes from eight men who had COVID-19. All eight testicular specimens showed no inflammation, vasculitis, vascular thrombosis, or morphologic evidence of viral changes. Viral nucleic acid was detected in one of the eight patients by RT-PCR [153]. Similarly, in the autopsy of a 77-year-old man who died of COVID-19 six days after the onset of symptoms, Barton reported that the testes were pathologically normal [68]. Bian reported that the testicular autopsy specimens showed various degrees of reduction and damage to the spermatogenic cells. The virus was also detectable in other tissues of the testis: reverse transcription polymerase chain reaction (RT-PCR), transmission electron microscopy (TEM), and immunohistochemistry (IHC) [151]. Enikeev et al. examined testes from autopsies of patients (*n* = 20) who had died from COVID-19. The RT-PCR test for viral nucleic acid was positive in all 20 patients. Pathologic findings showed that most specimens had structural disruption of testicular tissue, with evidence of damage to Leydig cells, germ cells, and microvascular endothelium combined with microthrombi. The stromal edema associated with lymphoid cell elements and the detection of SARS-CoV-2 viral proteins in spermatogenic epithelial cells, Leydig cells, and endothelial cells of blood vessels may indicate the viral nature of the injury [69]. Most studies report signs of orchitis, vascular changes, basement membrane thickening, damage to Leydig and Sertoli cells, and decreased spermatogenesis associated with SARS-CoV-2. There is disagreement about detecting the virus by RT-PCR, IHC, and TEM, with some studies reporting positive and negative results. The positive results for detecting the virus by RT-PCR may not be reliable because the technique may detect the virus in blood vessels rather than in testicular tissue. These results may be closely related to impaired hormone function and fertility in men with severe SARS-CoV-2 infection.

Fan et al. sought to determine whether the COVID-19 disease also affects male fertility. The researchers discovered that cells in the renal tubule, Leydig’s cells in the testis, and vas deferens had significant amounts of the ACE-2 receptor. This suggests the virus can infect and damage these reproductive cells and the kidneys [70]. These results highlight the need to screen young male patients for testicular tissue injury. For this reason, the study of SARS-CoV-2 and its effects on sperm is important.

On the other hand, we addressed the evidence that SARS-CoV-2 infection can cause infertility. For example, a study of sperm from 43 individuals showed that 18.6% were azoospermic; 7.0% had a sperm count of 2 million or less per milliliter; and 25.6% were either oligo-, crypto-, or azoospermic. In this study, oligo-, crypto-, or azoospermia was present in 25% of men who eventually recovered entirely from COVID-19. The authors suggested that these patients developed azoospermia after treatment with antiviral drugs, corticosteroids, antibiotics, chloroquine, or immunomodulators [18]. In addition, Erbayand et al. reported the results of sperm tests performed both before and after infection with SARS-CoV-2. Moderately symptomatic individuals had significantly lower total viability and progressive motility results. In the moderately symptomatic group, overall sperm parameters, including sperm volume, were reduced considerably [71]. Twenty-four male COVID-19 survivors were also analyzed by Pazir et al. [72]. Both total sperm motility and total sperm volume were statistically significantly reduced after COVID-19.

Moreover, after SARS-CoV-2 infection, even those who did not have fever due to COVID-19 disease had significantly lower sperm concentration and total motility [72]. Some antiviral drugs, including lopinavir/ritonavir and hydroxychloroquine, can affect sperm quality. Indeed, oxidative damage is induced in the testes of rats after lopinavir/ritonavir combination therapy, leading to changes in sperm characteristics and testicular integrity [22].

### 3.4. Hormonal Imbalance

It is suspected that SARS-CoV-2 causes low levels of LH, follicle-stimulating hormone (FSH), and T by triggering inflammatory responses that impair the normal function of the hypothalamic–pituitary–testicular (HPT) axis [134,152]. Studies comparing sex hormone levels in COVID-19 patients and healthy controls, including T, LH, and FSH, provide the only direct evidence of the effects of COVID-19 on male reproductive function. In COVID-19 patients, the ratio of T to LH and FSH to LH was significantly reduced, but T levels were comparable [66]. This may be the first definitive evidence that COVID-19 affects testicular sex hormone production. Still, it should be followed up with a more in-depth study of seminal fluid from COVID-19 patients to determine the effects of the virus on sperm quality [134,152]. There are discrepancies with this notion, however, as recent reports have shown that COVID-19 patients have lower blood T levels, higher LH levels, and lower T to LH ratio than healthy men [66].

COVID-19 may cause acute-stage hypogonadism [150] in male patients during the inflammatory phase, which may be responsible for exaggerated inflammatory responses and overproduction of cytokines [23,147]. T is essential as an immunosuppressor and cytokine regulator during inflammation [24]. Several studies have investigated the effects of COVID-19 on male sex hormone parameters in the acute phase of the disease. Rastrelli et al. [54] and Camici et al. [73] found that lower total testosterone (TT) and calculated free testosterone (cFT) were associated with worse clinical outcomes (intensive care unit (ICU) admission, death). In addition, higher LH was associated with worse disease severity. Low TT and cFT levels were significantly related to higher ferritin, procalcitonin (PCT), and LDH; elevated neutrophil levels; and decreased lymphocyte levels. In a prospective cohort study by Cayan et al., the more severe the disease, the lower the mean TT levels at admission and the higher the mean FSH and LH levels at admission [74].

COVID-19 patients have significantly lower TT levels and higher LH levels compared with healthy men, indicating primary hypogonadism [75]. This is supported by the results of pathologic testicular examination of men with COVID-19, who were found to have orchitis, vascular changes, basement membrane thickening, Leydig and Sertoli cell damage, and decreased spermatogenesis associated with SARS-CoV-2 [52,141]. SARS-CoV-2, after binding to neurons and glial cells expressing ACE-2, can cause neuroinflammation that can alter hormone balance and temperature regulation controlled by the hypothalamus [25]. As a result, there are concerns about the alteration of sex hormones, male reproductive organ health, and the ability of the testes to overcome the potential damage.

Several studies examined sex hormone levels in COVID-19 patients during recovery and compared them to sex hormone levels during hospitalization or before infection. A study by Karkin et al. found that TT levels were higher and LH levels were lower before COVID-19, while the difference in FSH levels was not statistically significant [76]. A prospective longitudinal study by Afshari et al. found that moderately and severely ill male patients had high LH levels and decreased TT [77]. Sex hormone levels of COVID-19 patients had more elevated prolactin and lower progesterone levels than those of the control group at initial sampling but normalized three months after discharge. Salonia et al. [78] noted significantly lower estradiol (E2) and LH levels and higher TT levels at follow-up. This could be explained by the physiological recovery of the damaged testes during the recovery period.

### 3.5. COVID-19 Vaccination and Male Reproductive Potential

Few studies have examined the effects of vaccination COVID-19 on male reproductive variables, such as sperm parameters, hormone levels, and testicular damage. Researchers have found no adverse effects of COVID-19 immunization on male fertility in the scientific literature. Abd et al. assessed the effect of (Pfizer-BioNTech, Mainz, Germany) mRNA COVID-19 vaccine on semen quality. Semen analysis of 60 men with normal semen parameters was compared before and after vaccination (at least 90 days after the two vaccine doses). No significant difference was found in semen volume, concentration, and morphology, except in total sperm motility (TSM) and progressive motility. The difference was clinically insignificant and within WHO normal ranges [79]. In the same line, Diaz et al. found no significant change in semen parameters (volume, concentration, total motility, total motile sperm count (TMSC)) between the baseline sample and three- and nine-month samples after vaccination in 12 healthy men who had completed the second dose of vaccination with mRNA vaccine (Pfizer, Moderna) against SARS-CoV-2. Two of three oligospermic men at baseline became normospermic, and one remained oligospermic in the follow-up. These findings suggest no long-term effects of mRNA COVID-19 vaccination [80]. Consistent with these results, another study with a short follow-up examination (one spermatogenesis cycle) by Gonzalez et al. included 45 healthy men. Interestingly it was found that semen volume, concentration, motility, and TMSC were significantly increased in the follow-up sample obtained approximately 70 days after the second dose. This increase may be due to higher abstinence time in the follow-up sample. Samples of men with oligospermia did not show further decline [81]. Barda et al. observed no deleterious effect of (Pfizer) mRNA vaccine on sperm quality for fresh and frozen-thawed samples of 33 sperm donors. Multiple semen samples (*n* = 898 in total) were collected before and after the vaccination, and at least one sample was collected 75 days after the second dose. Interestingly total sperm count (TSC) and TMSC increased after vaccination. Moreover, the number of samples available for freezing was higher, and the presence of motile sperm after freezing was higher post-vaccination, suggesting the vaccine’s safety in the long term after vaccination [82]. In contrast, Gat et al. observed a deterioration of sperm concentration and total motile count (TMC) in semen analysis three months after vaccination with two doses of (Pfizer) mRNA vaccine in 37 sperm donors who had provided pre- and post-vaccination samples three different times. The overall recovery of semen parameters without abnormalities was observed in the following samples. The authors suggested that the transient impairment results from the febrile systemic response, often observed after vaccination [83]. Karavani et al. evaluated semen analysis of 58 men undergoing in vitro fertilization (IVF) treatments who received at least two doses of (Pfizer) mRNA vaccine and submitted semen samples before and 6 to 14 months after vaccination. Semen parameters (volume, concentration, motility, morphology, TMC) for pre- and post-vaccination samples were comparable. In addition, the subgroup of men who received a third dose and subgroups of men with normal and abnormal semen parameters pre-vaccination did not present any difference. The authors suggested the safety of the mRNA vaccine in the long-term follow-up [84]. The same results were noted in a similar study by Safrai et al. of 72 men undergoing infertility treatments who provided semen specimens before vaccination and 71 days after their two doses [85]. Olana et al. found no difference in semen parameters (concentration, motility, and morphology) of 47 men before and after vaccination with the (Pfizer) mRNA vaccine. The follow-up semen analysis was contacted three months after the two doses. Furthermore, no difference was observed in oxidative stress analysis, IL-6 (as a marker of inflammation), and electrolyte function pre- and post-vaccination [86]. It is evident that mRNA vaccines do not exert long-term negative effects on semen parameters in men with both normozoospermic and abnormal semen profiles before vaccination. Many authors have suggested that mRNA vaccines are unlikely to affect sperm quality because they do not contain the live virus. Elhabak et al. studied the effect on the semen quality of inactivated virus (Sinopharm, Beijing, China) and viral vector (AstraZeneca, Cambridge, UK) vaccine. Semen samples of 100 healthy men were collected before vaccination and 70 days after the second dose. No significant change was found between semen parameters pre- and post-vaccination in both vaccines [87]. Similar results were reported by Zhu et al. No significant change in semen parameters (volume, concentration, progressive motility, total progressive motile sperm count (TPMC)) before vaccination and after the second dose of inactivated virus vaccine in 43 sperm donors was found [88]. Meitei et al. compared semen parameters in a cohort of 53 sub-fertile men before and after the viral vector vaccine (Covishield, Pune, India). No significant difference was observed in men (both with normal or abnormal semen analysis before vaccination) before and after vaccine administration, except morphology, which showed a moderate decline, probably clinically insignificant as it was within WHO reference values. Follow-up semen samples were obtained approximately 83 days after the second dose [89]. Until now, there have been no concerns regarding the safety of inactivated virus and viral vector vaccines on semen quality based on current literature published. Carto et al. studied testicular tissue responses to immunization against COVID-19. Vaccinated and unvaccinated subjects were evaluated for orchitis and epididymitis. The results suggested that a single dose of the vaccine COVID-19 could significantly reduce the risk of testicular inflammation, as rates of orchitis and epididymitis were significantly lower in vaccinated patients compared with unvaccinated patients [145].

Orvieto et al. investigated the effect of the mRNA COVID-19 vaccine on 32 couples undergoing IVF treatments. No difference was found in fertilization rate, pregnancy rate, and number of oocytes retrieved before and after vaccination. In addition, semen quality did not change after immunization [90]. In the same line, Reschini et al. showed that COVID-19 vaccination with mRNA or viral vector vaccine did not affect the fertilization rate and sperm analysis of 106 men assisted with assisted reproduction technology (ART) before and after vaccination. The median time between the first vaccine dose and the second ART cycle was 75 days [91]. A study by Xia et al. compared semen parameters and embryo quality between two groups. One group consisted of 105 men who had completed two doses of COVID-19 inactivated vaccine [(Sinovac, Beijing, China), (Sinopharm, Beijing, China)]. The other consisted of 155 unvaccinated cases; men of both groups participated in IVF treatments. Semen parameters (volume, concentration, motility) were similar between the two groups.

Moreover, no significant difference was detected in IVF outcomes (embryonic development, blastocyst quality) and pregnancy rates (biochemical, clinical) in both groups [92]. Similar results were reported in a prospective cohort study by Wang et al. that enrolled 542 couples undergoing infertility treatments and divided them according to the history of vaccination of male partner into two groups, the vaccinated group (*n* = 275) and the unvaccinated group (*n* = 944) [93]. COVID-19 vaccination was not associated with altered IVF outcomes, so it should be recommended to men seeking fertility. Currently, no evidence supports the defect of male fertility by COVID-19 vaccination. COVID-19 infection can injure the testes and alter the semen quality, while the vaccine may help prevent the illness and maintain and protect reproductive potential, especially in subfertile men [148].

Although some individuals may experience mild fever after the second dose of COVID-19 vaccination, this does not pose a risk to the male reproductive system, as recently reported [26]. Therefore, immunization did not induce a robust inflammatory response throughout the reproductive system. In addition, there is no evidence that vaccination damages testicular tissue or affects sperm quality. Table 3 presents a summary of studies that have examined the impact of antioxidant supplementation on male infertility.

### 3.6. Oxidative Stress and Sperm DNA Fragmentation

SARS-CoV-2 pathophysiology involves inflammation, cytokine overproduction, and cell death. These processes are associated with reactive oxygen species (ROS) overproduction and induction of OS. High ROS concentration during the infection activates the nuclear factor kappa-light-chain-enhancer (NF-kB) of the activated B-cell Toll-like receptor (mainly TLR-4). This pathway increases cytokine release, exaggerating the inflammatory response [8]. Moreover, inhibition of nuclear-factor-erythroid-2-related factor 2 (Nrf-2) could be another pathway exploited by SARS-CoV-2, leading to inflammation and oxidative stress [19]. Both oxidative stress and inflammatory cytokines have been reported to damage testicular cells. In particular, oxidative stress can damage Leydig cells, disrupting testosterone production and damaging germ cells, altering spermatogenesis [27]. At the micro level, OS can defect male fertility predominantly through sperm DNA fragmentation, sperm membrane lipid peroxidation, and induction of apoptosis in the spermatozoa [135]. The following studies have produced essential data supporting OS as a potential mechanism for the testicular damage observed in COVID-19 patients (Figure 2).

In a prospective longitudinal cohort study, Maleki et al. investigated the connection between reproductive function and alterations in multiple semen biomarkers in men recovering from COVID-19. Semen parameters, inflammatory and oxidative stress markers, apoptotic variables, and ACE-2 seminal activity were evaluated in COVID-19 patients (*n* = 84) with different severity infections and proven fertility at the point of hospital discharge and every ten days until two months were completed and were compared with healthy controls (*n* = 105). The COVID-19 group showed significantly higher ROS levels, pro- and anti-inflammatory cytokines (IL-1β, IL6, IL-8, TNF-α, transforming growth factor-β (TGF-β), interferon (IFN-α, IFN-γ)), apoptotic variables (TUNNEL percentage; caspase-3, caspase-8, and caspase-9 activity), and semen ACE-2 enzymatic activity, as well as significantly lower superoxide dismutase (SOD) activity. These alterations persisted over time; their magnitude was related to disease severity and was associated with defective semen quality (semen volume, concentration, spermatozoa number, progressive motility, and morphology). The authors suggested that ROS overproduction and alteration of antioxidant machinery are related to excessive pro- and anti-inflammatory responses and cause apoptosis, cytopathological alterations, and DNA damage in sperm cells, thus creating a transient state of male subfertility [94].

Similarly, Falahieh et al., in a prospective cohort study, examined the effect of COVID-19 infection on oxidative stress markers and semen parameters of 20 fertile men hospitalized with moderate disease in semen samples obtained 14 and 120 days after the COVID-19 diagnosis. It was observed that sperm morphology and total and progressive motility were below the normal range values recommended by the WHO, and peroxidase-positive leucocytes and DNA fragmentation index (DFI) percentage were above normal range values. However, all the parameters improved significantly after 120 days and returned to reference range values, except morphology, which remained below them. Of note, ROS concentration and malondialdehyde (MDA) in semen samples obtained after 120 days significantly decreased compared to 14 days, and semen total antioxidant capacity (TAC) significantly increased. It was suggested that systemic inflammation during disease is associated with high peroxidase-positive leucocytes and high ROS concentrations in seminal fluid, causing detrimental effects on sperm properties, which tend to subside after 120 days [95].

Similarly, a case report study by Charagozloo et al. reported that a man moderately infected by COVID-19, normozoospermic before the infection, presented azoospermia during the disease that persisted for four weeks. Spermatogenesis recovered subsequently and was accompanied by high levels of oxidative DNA damage [154]. Another study by Shcherbitskaia et al. compared semen parameters, levels of sperm DNA fragmentation, and pro-antioxidant markers in semen samples of 17 men five months after SARS-CoV-2 infection, with 22 men never infected. It was observed that DNA fragmentation level was negatively associated with the recovery period after infection. It was also detected that COVID-19 was not always correlated with DNA fragmentation levels. Thus, the authors considered them as two independent factors. COVID-19 patients with abnormal TUNNEL rate showed the most significant alterations, such as decreased seminal fluid nitrotyrosine levels (NT), TAC, and zinc (Zn), as well as increased 8-hydro-2-deoxyguanosine (8-OHdG) levels within spermatozoa. Furthermore, increased round cell numbers in semen were observed. The authors suggested that dysregulation of pro-antioxidant balance in semen after COVID-19 can cause an increase in DNA fragmentation and defects in semen quality [96].

Moghimi et al. investigated the incidence of apoptosis in testis tissue obtained from autopsies of deceased COVID-19 men (*n* = 6) and non-COVID-19 diseased men (*n* = 6). The COVID-19 group showed a significant decrease in the seminiferous tubules and interstitial tissue volume. It was also observed that the reduction of testicular cells (spermatogonia, primary spermatocytes, spermatids, Sertoli and Leydig cells) indicated testicular damage. Examination of caspase 3, Bcl-2-associated X protein (BAX), B-cell lymphoma protein (BCL) expression, and TUNNEL assay showed increased apoptosis of testicular cells. More importantly, significant increase in ROS production and glutathione (GSH) concentration reduction was observed compared to the control group. These results suggest that COVID-19 induces apoptosis in testicular cells through redox balance dysregulation, associated with COVID-19 pathogenesis via blood clotting, hypoxia induction, and cytokine storm exaggeration. In addition, cytokine storm and oxidative stress are related to acute stage hypogonadism in COVID-19 patients, posing an extra danger for male fertility [97].

Similarly, Flaifel et al. analyzed testis and epididymis specimens of ten autopsies from patients who succumbed to COVID-19. The examination revealed morphological alterations that could be attributed to oxidative stress (chromatin condensation, nuclear fragmentation, and acidophilic cytoplasm). Sloughing of spermatocytes into the tubular lumen and accumulation in the epididymis head, spermatids elongation, swelling, and vacuolation of Sertoli cells were also observed. In addition, multifocal microthrombi, increased platelets in testicular vessels, and increased mononuclear inflammatory infiltrates (CD8 positive) in the interstitial, compatible with orchitis, were detected. As suggested by the authors, these findings are evidence of acute testicular injury related to oxidative stress. Moreover, coagulation dysregulation leading to microthrombi, toxic metabolic effects of prolonged disease, and cytokine storm could be contributing factors for these alterations [133] (Figure 2).

In addition to the direct association of SARS-CoV-2 with OS, treatments administered during the disease, such as antiviral treatments, are also related to ROS production and sperm DNA fragmentation [27]. High fever is a common symptom of COVID-19. Testicular heat is reported that can induce OS as well as sperm DNA fragmentation [28,98]. As noted by testicular tissue examination, SARS-CoV-2 can cause orchitis and influence OS. Moreover, COVID-19 is often associated with psychological stress, a major cause of systemic OS [152]. These findings emphasize that oxidative stress increases and antioxidant defenses are reduced during COVID-19. Hence, we could hypothesize that antioxidants could be a beneficial adjuvant therapy for COVID-19 disease to protect testicular function and ensure proper steroidogenesis and spermatogenesis [29,30] (Figure 2). Table 4 provides an overview of the studies that have explored the effects of COVID-19 on seminal biomarkers and testicular health.

## 4. Antioxidants and Male Infertility—What Is Currently Known?

Oxidative stress is a common feature of many factors and conditions that negatively affect male fertility. The current literature reports that OS contributes to 80% of male infertility [31]. Reactive oxygen species are essential in the physiological mechanisms of male fertility [99]. A small amount of ROS has a significant role in sperm capacitation, acrosome reaction, and hyperactivation processes [32]. However, ROS overproduction and OS cause many detrimental effects on sperm, as highlighted. The function and structure of spermatozoa are affected mainly by OS; their plasma membrane is rich in polyunsaturated fatty acids, making them very vulnerable to lipid peroxidation. In addition, the spermatozoa nucleus and mitochondria epigenome are also vulnerable to high ROS levels, causing DNA damage via DNA fragmentation, microdeletions, and mutations [32]. Clinically OS translates to decreased sperm quality and can lead to declined fertilization, poor embryonic development, recurrent pregnancy losses, genetically heritable mutations, and overall poor ART outcomes [31]. The human body can maintain a balanced oxidative environment, regulated through various enzymatic and non-enzymatic defense mechanisms, further classified as endogenous and exogenous antioxidants [99]. The most essential endogenously produced enzymatic antioxidants include catalase (CAT), superoxide dismutase (SOD), and glutathione peroxidase (GPX). Non-enzymatic antioxidants are endogenously produced or exogenous received through food or supplements. The most important naturally non-enzymatic antioxidants constitute glutathione, ascorbic acid (vitamin C), tocopherols (vitamin E), carotenoid (Lycopene), coenzyme Q10, and trace metals (zinc, selenium) [99]. Antioxidant regimens have been used in the last decades as a quick, relatively safe, inexpensive solution to confront seminal OS and male infertility. Current evidence reports a positive association between antioxidant supplements administration and sperm quality improvement, thus presenting a promising treatment.

Given the valuable properties of antioxidants to counteract OS, physicians often prescribe antioxidant supplements to treat or protect patients from oxidative-stress-related disorders, including male infertility cases. In addition, it is a common practice for many people to consume over-the-counter antioxidant regimens as promoters of good health to combat diseases in healthy living. Moreover, in the absence of a commonly accepted assay for detecting OS, antioxidants are often prescribed or used by patients indiscreetly without prior screening for OS existence [136]. As a result, many patients can take high doses of antioxidants. Excessive use of antioxidants may lead to adverse effects on male fertility altering redox balance and inducing oxidative stress [33]. Halliwell et al. reported this paradoxical phenomenon and termed it an “antioxidant paradox” [156]. A delicate balance between oxidant and reductive agents is essential to maintain redox equilibrium. Uncontrolled, high exposure to reductants (antioxidants) may induce a shift in a reduced state called “reductive stress”, with subsequent detrimental effects on cell function, such as oxidative stress. Some of these adverse effects reported in the existing literature include cardiomyopathy, blood–brain barrier dysfunction, cancer, and embryogenesis defects [33]. Verma’s in vitro analysis study assessed semen parameters after ascorbic acid supplementation to semen samples. It was observed that concentrations below 1000 μM increased sperm motility and decreased lipid peroxidation. However, concentrations above 1000 μM caused the complete reverse effect, suggesting that concentrations above 1000 μΜ are not protective for semen but rather harmful [100]. Menezo et al. assessed the DNA fragmentation index and the degree of DNA decondensation to 58 infertile men before and after treatment with antioxidants (vitamin C, vitamin E, β-carotene, zinc, and selenium) at specific doses. It was observed that DNA fragmentation decreased and DNA decondensation increased after 90 days of treatment. A possible explanation for this adverse effect may be the property of vitamin C to break interchain disulphide bonds of protamine in sperm DNA, thus defecting paternal gene activity during preimplantation development [101]. Bleau et al. studied the association between selenium (Se) concentrations in semen and semen quality of 125 men from couples consulted for infertility. It was detected that Se levels between 40 to 70 ng/mL were optimal for reproductive performance. However, Se levels below this range were associated with infertility, and above this range were associated with a high abortion rate, decreased motility, and asthenospermia [102]. These studies make evident the paradoxical effects of antioxidants on male fertility, which seem dose dependent. Therefore, antioxidants should be cautiously administered before redox balance examination, and these patients should be carefully followed up [33].

Several different studies have reported a positive impact of antioxidants on semen quality. A placebo-controlled, double-blind, randomized trial by Balercia et al. assessed the improvement of semen quality in 60 infertile men (idiopathic asthenospermia) treated with exogenous coenzyme Q10 (CoQ10) after six months of therapy. It was observed that levels of CoQ10 and ubiquinol were increased in semen, and sperm kinetic features were improved after treatment. Moreover, in the treated group, six pregnancies were observed, twice as many as those observed in the control group. Patients with lower baseline levels of CoQ10 and lower sperm motility had greater chances of being favored by the treatment [103]. The same group evaluated the effectiveness of single treatment with L-acetyl-carnitine (LAC) or L-carnitine (LC) and combination treatment with LAC and LC in improving semen kinetic parameters and total scavenging capacity (TOSC). This placebo-controlled, double-blind, randomized trial included 60 men with idiopathic asthenospermia supplemented for six months. Sperm cell motility improvement was detected in patients administered with LAC single treatment or in combination with LC.

Furthermore, the TOSC assay significantly improved hydroxyl and peroxyl radicals for treated patients. Interestingly, nine pregnancies were achieved during the therapy in the treated group. Patients with lower baseline motility and higher TOSC values were also observed to have a significantly greater probability of after-treatment improvement [104]. A monocentric, randomized, double-blind, placebo-controlled trial by Busetto et al. studied the effect of supplementation with an antioxidant formula (LC, LAC, fructose, fumarate, vitamin C, vitamin B12 citric acid, zinc, selenium, CoQ10) on semen quality after six months. Ninety-four patients with oligo- and/or astheno- and/or teratozoospermia were included. Sperm concentration, total sperm count, and total and progressive motility were significantly increased in the supplemented group compared to the placebo group. Although the pregnancy was not an endpoint, ten pregnancies were recorded during the follow-up period in the treated group, while two were recorded in the placebo group. These improvements were more evident in the group of patients with varicocele [105].

Another randomized control trial by Tremmelen et al. investigated the effect of male antioxidant supplementation on pregnancy rates and embryo quality. Sixty couples undergoing in vitro fertilization (IVF)–intracytoplasmic sperm injection (ICSI) treatments with male factor infertility were enrolled in the study. Male partners were administered Menevit, an antioxidant consisting of vitamins C and E, selenium, garlic lycopene, zinc, and folic acid, for three months before IVF-ICSI treatments. Statistically significant improvement in viable pregnancy rate was found in the antioxidant-administered group compared to the placebo group. No statistically significant improvement was detected in the two groups’ oocyte fertilization and embryo quality rates [106]. A systematic review by Salvio et al. studied the effectiveness of CoQ10 alone or in combination with other antioxidants as a treatment for male infertility—an improvement in sperm quality, especially sperm motility, was noted after CoQ10 administration. Most improvements were observed 3 to 6 months following treatment initiation and perished after treatment discontinuation [123].

Similarly, a systematic review by Tsampoukas et al. reported that semen parameters of men with varicocele-related infertility were improved after antioxidant supplementation (vitamins in combination with other molecules) and could be used as primary or adjuvant therapy [124]. In contrast to these findings, Greco et al. and Rolf et al. failed to detect any improvement in the semen quality of infertile men after supplementation with antioxidant regimens (vitamin C, vitamin E) [107,108]. Along the same line, a systematic review and meta-analysis by Pyrgidis et al. included 14 studies. Antioxidant regimen supplementation did not seem to improve pregnancy rates, DNA fragmentation, and semen parameters in patients with varicocele-related infertility. More specifically, it was reported that patients with surgical treatment showed no significant differences after 3 and 6 months of antioxidant treatment without pre-treatment OS screening [126]. Although numerous direct and indirect diagnostic techniques for measuring OS are available, neither succeeded as a technique used in clinical practice due to methodological complexity, time-consuming procedures, high cost, and need for extensive sample volume [31]. Oxidation-reduction potential (ORP) reflects the balance between oxidants and reductants (antioxidants) and constitutes an essential measure of redox balance, especially OS. The MiOXSYS System has emerged recently as a reliable and clinically useful method of measuring ORP, providing a practical way to assess OS that clinicians should take advantage of [31].

However, the role of antioxidant treatment in male infertility is still uncertain. A Cochrane systematic review and meta-analysis showed that antioxidant therapy positively affected the number of pregnancies and live births in sub-fertile couples undergoing ART treatment [127]. Thirty-four randomized control trials (RCTs) with 2.876 couples who were administered different antioxidant regimens were included in this study [127]. Similar results were reported in a recent metanalysis, which included 61 trials and 6264 sub-fertile men [128]. A recent prospective study that assessed the impact of oral antioxidants (consisting of coenzyme Q10, omega-3, and oligo-elements) and lifestyle changes on semen quality after three months reported similar results. The study enrolled 93 men who had undergone IVF and ICSI attempts and ten men who were used as controls. No differences in semen quality and static oxidation reduction potential (sORP) were observed [109].

The Males, Antioxidants, and Infertility (MOXI) study was a multicenter, double-blind, randomized controlled trial from 2015 to 2018. A total of 174 men were enrolled and divided into two groups, the study group (*n* = 85) that received an antioxidant supplement (vitamins C, E, selenium, L-carnitine, zinc, folic acid, and lycopene) and the placebo group (*n* = 86). After six months, no significant difference was observed in the cumulative live birth rate among the two groups. An interval pilot study that was carried out on these patients aimed to assess the semen parameter and DFI changes three months after the supplementation initiation; no improvement was observed [157]. A later study tried to determine the role of serum levels of antioxidants (vitamin E, zinc, or selenium) on semen quality and birth rate. No association between those factors was revealed [110].

Antioxidant administration remains a significant challenge, so unyielding evidence about their therapeutic value is lacking [34]. Some studies present consistent benefits from antioxidant supplementation, though these data are controversial and are not commonly accepted, as emphasized by many reviews [35]. Dimitriadis et al. pointed out that over-the-counter antioxidant use could harm male fertility [32]. Moreover, only small cohorts have been used for relevant analyses, indicating that current evidence is of low quality. For this reason, European Guidelines on Sexual and Reproductive Health for Male Infertility does not recommend using antioxidants so far [36].

Eight RCTs have been published on the effect of antioxidants on semen quality (six were placebo-controlled) in the last two years. Although some antioxidant combinations positively impacted semen quality, the differences were generally minor, and no significant change in pregnancy rates was detected, probably due to their small sample size. L-carnitine, N-acetylcysteine (NAC) [111,112,113,129,130], coenzyme Q10 [114,115], and vitamin D [116] were some of the antioxidant combinations used in these studies with a positive impact on semen quality.

However, many significant limitations are present in the design of these studies. Data were drawn from poor-quality RCTs, with inadequate methods of randomization reporting, increasing the risk of bias, as well as studies lacking data about clinical outcomes such as clinical pregnancy and live birth rates. As a result, conclusive evidence on which antioxidants should be used to improve semen quality and pregnancy rates has not been reached [128].

Antioxidants are indeed an up-and-coming solution in OS-induced infertility. However, they should not be administered indiscreetly to infertile patients. Before antioxidant administration, OS screening should be performed on all patients to determine who is more likely to be favored. In this way, we can protect the patients without the need for antioxidant supplementation from the adverse effects of high exposure to antioxidants. Future research should provide dosing guidelines, administration period, and an optimal mixture of antioxidants for more effective results and safer usage of these regimens [32]. A comprehensive overview of studies that assess the effectiveness of antioxidant supplementation on male infertility is provided in Table 5.

## 5. Could Antioxidants Play a Role in Counteracting COVID-19 OS-Induced Damage in Male Reproduction?

A plethora of evidence indicates that OS is strongly involved in spermatogenesis alteration observed in COVID-19 patients after the infection. Although the ability of many antioxidants to ameliorate semen quality and improve semen redox balance is documented, the potential of antioxidant use in men after SARS-CoV-2 to support male fertility is under-researched. A few studies suggest that supplementing micronutrients with antioxidant potential in men after SARS-CoV-2 infection may accelerate the recovery of altered semen parameters and decrease seminal OS induced by COVID-19 disease, thus protecting male fertility. An interventional study by Rafiee et al. investigated the effect of antioxidant N-acetylcysteine (NAC) on abnormal semen parameters in men infected by COVID-19 before and after NAC supplementation. The study included 200 men referred to infertility clinics due to female factor infertility and was diseased the last two months caused by SARS-CoV-2. Men were divided equally into two groups: the supplemented group (*n* = 100), which was administered 600 mg/day per os for three months, and the control group (*n* = 100). Semen samples obtained after COVID-19 infection on an average of 6 weeks (3 to 8 weeks) presented significantly decreased semen parameters (sperm concentration, motility, and normal morphology) compared to pre-COVID-19 semen analysis. The follow-up semen analysis three months after NAC treatment initiation showed that semen parameters significantly improved and were comparable to the pre-COVID-19 infection values. In contrast, the three months follow-up semen analysis in the control group was impaired, and semen parameters were decreased compared to the values before COVID-19. It was indicated that NAC administration may positively affect semen quality [117]. Another study by Kurashova et al. examined the efficacy of an antioxidant regimen on men with pathozoospermia after COVID-19 infected them [118]. The antioxidant complex consisted of astaxanthin and omega-3 fatty acids and was administered to patients thrice daily over one month. Blood and semen samples were collected during the recovery period approximately two months after negative PCR results, and follow-up samples were collected after the completion of the antioxidant treatment (one month). COVID-19-infected men presented a significant decrease in sperm concentration, motility, and viability compared to pre-disease values. In addition, leucocyte numbers were significantly increased. After the antioxidant treatment, semen parameters improved and were comparable to the reference values, and the leucocyte number significantly decreased. Moreover, patients during the recovery period before the treatment showed increased TBA reactive substances (TBARs) and decreased total antioxidant activity (AOA) levels in the blood samples compared to controls. After the complex antioxidant administration, TBARs and AOA levels were significantly improved. It was suggested that the antioxidant treatment contributed to restoring the balance in the lipid peroxidation, antioxidant defense system, and normalization of semen parameters [118]. A case report study by Mannur et al. reported that a normozoospermic man undergoing infertility treatments due to female fertility factor developed oligo-astheno-teratozospermia with severe sperm DNA damage one month after his recovery from COVID-19 infection with mild symptoms. The patient was treated with antioxidant and multivitamin supplements consisting of coenzyme Q10, lycopene, docosahexaenoic acid, folic acid, selenium, and zinc for three months. The semen analysis after the treatment showed a significant improvement in sperm count and motility, but morphology and sperm DNA damage remained defective [155]. In the same line, Aschauer et al. studied the efficacy of micronutrient supplementation to improve seminal OS and semen variables after symptomatic SARS-CoV-2 infection in comparison to SARS-CoV-2-infected men without supplement intake. This prospective comparative study enrolled 30 men without subfertility history who provided semen samples not more than 12 weeks after a positive SARS-CoV-2 PCR test. They received daily a micronutrient supplement consisting of L-carnitine, L-arginine, zinc, vitamin E, glutathione, selenium, coenzyme Q10, and folic acid (Profertil) by oral route for 12 weeks. After three months of supplementation, the number of cases with semen parameter values within WHO reference limits increased significantly in the study group. It also observed a significant increase in vitality, as well as in total and progressive motility. The control group showed no significant improvement in semen parameters and sORP levels three months after the initial semen analysis. The sORP levels in the supplemented group showed an evident decrease 12 weeks after the treatment initiation, although not statistically significant. The participants reported no serious adverse effects administered the supplement. The authors suggested that antioxidants may be protective against ROS overproduction in the seminal fluid. The studies mentioned above indicate that antioxidants could be a simple and safe yet effective way to support the recovery of semen quality after COVID-19 infection (Table 1) and optimize the reproductive potential of infected men currently trying for children, especially those men undergoing ART treatments and those that do not have time as an ally [119].

Since numerous oxidation-sensitive mechanisms are involved in COVID-19-mediated male infertility, OS-targeted therapies may effectively address the problem [37]. There is evidence that early administration of a high dose of vitamin C may reverse these unfavorable outcomes. Vitamin C is essential to the body’s natural antioxidant defenses; it was found to be helpful when administered in critical care [38]. Clinical studies in 146 individuals with sepsis suggested that high doses of intravenously administered vitamin C may be an effective therapy. Evidence showed that treatment with vitamin C and sulforaphane could attenuate the acute inflammatory lung injury caused by OS [37]. The testes highly depend on vitamin C, an antioxidant that is very efficient in counteracting ROS and reducing sperm clumping. In addition to protecting testicular cells from OS, it also helps minimize sperm DNA fragmentation by contributing electrons to redox systems, inhibiting lipid peroxidation, recycling vitamin E, and protecting DNA from damage by peroxide radicals [38]. Accordingly, men may benefit from taking vitamin C as part of their COVID-19 treatment plans. To determine whether vitamin C therapy, in addition to standard treatment, is effective in curing systemic viral infection and restoring male reproductive function in COVID-19 male patients, carefully designed clinical trials are needed. The severity of COVID-19 may be affected by selenium (Se) deficiency. A study in 17 Chinese cities found that the incidence of COVID-19 was positively correlated with endogenous Se concentration, and Se deficiency was associated with an increased risk of dying from COVID-19 [137]. The possible role of the micronutrients zinc and selenium in the development and progression of COVID-19 has not been adequately investigated. However, it has been speculated that administering nutritional supplements containing minerals in the early stages of infection may improve host resistance [39].

Ebselen is an organo-selenium molecule that, like glutathione peroxidase and peroxiredoxin, inhibits the production of hydroperoxide and peroxynitrite. The pleiotropic properties of ebselen, particularly its antibacterial, antiviral, and anti-inflammatory activity, can be attributed to its ability to form a selenosulfide bond with various thiols in proteins. The protease (Mpro) of the primary SARS CoV-2 is a promising therapeutic target. Ebselen is one of the potent Mpro inhibitors that have emerged from the study of more than 10,000 molecules [37]. Antioxidant activities mediated by the Se-dependent enzyme glutathione peroxidase (GPx1) are the primary function of Se in mammals. The GPx1 mechanism prevents peroxidative damage to membranes and organelles caused by OS. In addition, Se has been found to improve semen parameters in infertile males via this action mechanism and increase pregnancy rates [37]. The beneficial effects of Se on male fertility and COVID-19 via comparable processes lead some to believe that it may be a good choice for studies to restore male reproductive function in COVID-19-infected men.

Other well-established conditions include zinc deficiency and worse disease outcomes of respiratory viral infections. Intracellular zinc depletion could be another factor that causes alteration of various zinc-dependent antioxidant proteins such as GSH, SOD, catalase, nitric oxide (NO) synthase dimer assembly, and other zinc finger proteins, thus causing mitochondrial damage and oxidative stress induction. Moreover, zinc inhibits nicotinamide adenine dinucleotide phosphate (NADPH) oxidase and is an antagonist to the redox activity of transition metals (copper, iron), thus preventing ROS production. As a result, zinc depletion during COVID-19 may harm all organs, including the testis [40]. Zinc is anti-inflammatory, inhibits NF-kB pathways, and limits cytokine storm via T-cell function regulation. Thus, zinc administration during COVID-19 disease could be valuable in improving the host immune response and preventing ROS overproduction [40]. The current literature has scarce evidence about zinc supplementation to COVID-19 patients. A retrospective study showed no correlation between zinc supplementation, survival, and improved prognosis [138]. It is suggested that zinc should be administered in the pre-cytokine storm phase to replenish zinc levels in the body and potentially prevent COVID-19 progression by suppressing excessive immune reaction and preventing ROS overproduction. After the cytokine storm and OS rule over, zinc administration may be of limited value [40]. Zinc is also a crucial regulator of male reproductive function. It has an essential role in sperm membrane stabilization, acrosomal reaction, and capacitation, as well as a regulative role in the oxidative metabolism of sperm. Zinc deficiency has been correlated with testosterone deficiency, primary testicular damage, dysfunction of LH receptors, and OS-induced Leydig cell damage [40,120]. Given the essential role of zinc in male reproduction and its antioxidant and anti-inflammatory properties, zinc supplementation during and after COVID-19 disease may be beneficial in preventing disease progression and protecting male fertility.

N-Acetylcysteine (NAC), a potent antioxidant biomolecule, could combat ROS production and, more importantly, the “cytokine storm” seen in COVID-19 [41]. Pro-inflammatory cytokines such as IL-1, IL-2, IL-4, TNF-α, and interferons (IFNs) are thought to be involved in the immunological response triggered by SARS-CoV-2, as was the case with SARS-CoV-1. By downregulating signal transducer and activator transcription 1 (STAT1), SARS-CoV-1 infection blocks the IFN-stimulating effect of type I IFNs. A cytokine storm in SARS-CoV-2 infection inhibits the IFN response. It has been suggested that N-acetylcysteine may increase Toll-like receptor-7 (TLR-7) and mitochondrial antiviral protein signaling cascades, reviving type I IFN production in response to SARS-CoV-2 [41]. In the pathophysiology of SARS-CoV-2, NF-kB acts as a mediator by triggering a cytokine storm. However, NAC restored the thiol pools and ROS scavenging mechanism in an in vitro influenza A and B model by blocking NF-kB activation [41].

Inflammatory diseases progress more rapidly when GSH levels are low and ROS formation is high. According to a recent study, patients with severe COVID-19 infection had a higher ROS/GSH ratio than patients with mild disease [41]. Patients with moderate COVID-19 symptoms were treated with NAC administered orally, intravenously, or inhaled as a more cost-effective therapeutic option. Patients with COVID-19 who received 6 g of NAC intravenously daily fared better in a recent clinical trial [37]. Oral NAC (600 mg/day) may also protect against SARS-CoV-2 in individuals often exposed to the virus. NAC’s anti-inflammatory and antioxidant properties may protect tissues from oxidative damage, which can lead to male reproductive failure in COVID-19 patients due to secondary immunological responses triggered by systemic inflammation and OS [41]. Infertile men taking NAC have been shown to have improved oxidative/antioxidant status and sperm parameters [121,122]. NAC can degrade the cross-linked glycoprotein matrix of mucus due to its free sulfhydryl group. Since NAC can cleave viral disulfide bonds, it may also help prevent SARS-CoV-2 from entering testicular cells. As an essential substrate for glutathione synthesis, NAC is an effective anti-inflammatory and antioxidant agent that regulates redox by keeping the thiol pool stable [41]. NAC’s anti-inflammatory and antioxidant properties could support the protection of male reproductive processes from COVID-19-mediated damage. Therefore, investigating the potential efficacy of NAC in reducing COVID-19-mediated impairment of male reproductive function is an important area of research.

Astaxanthin is a photo-protective red pigment of the carotenoid family with powerful antioxidant and anti-inflammatory activity. Astaxanthin can balance the immune system by inhibiting unnecessary inflammation and reducing excessive immune responses [42]. There is evidence that astaxanthin has beneficial effects and protective properties as treatment in the prevention of various diseases such as cardiovascular diseases, diabetes, liver diseases, and chronic inflammatory diseases. This natural compound has an essential effect on the regulation of various inflammatory factors (IL-6, IL-8 IL-1B, TNF-α), affects the NF-kB signaling pathway, inhibits the cyclooxygenase (COX1 and 2) enzyme activity, and suppresses prostaglandin E-2 (PGE-2) and nitric oxide (NO) [42]. Astaxanthin is often used as a critical ingredient in dietary supplements for improving male fertility. As already demonstrated by many researchers, astaxanthin positively impacts the male reproductive system and semen quality by increasing linear velocity sperm concentration and sperm capacitation, thus ameliorating male fertility [139]. It is also reported that astaxanthin supplementation improves sperm membrane fluidity by the induction of changes in the phospholipid composition of the spermatozoon membrane [118]. Spermatogenesis alteration via inflammation and OS induction are common in many COVID-19 patients. In this condition, astaxanthin supplementation could help reduce inflammation and keep the essential levels of ROS required for sperm capacitation.

COVID-19 pathophysiology is closely related to OS and semen quality deterioration observed in men after COVID-19 [43]. Antioxidants seem to be a promising adjuvant treatment against the disease progression with a simultaneous protective role to male fertility. Further research should be carried out and shed light on the abilities of these treatments to address both COVID-19 and its impacts on male reproductive functions.

Several seminal fluid proteins have been linked with increased levels of oxidative stress so far. Analyzing differentially expressed proteins in the normal and abnormal semen samples is a crucial step in deciphering critical molecular pathways of idiopathic infertility. Biomarkers such as DJ-1, prolactin-inducing protein (PIP), and lactoferrin have been investigated in a handful of proteomic studies that sought to understand the basis of oxidative-stress-induced male infertility. Undoubtedly, proteomics is a challenging field of increasing interest, and proteomic analyses warrant further research to improve our collective understanding of male infertility at a molecular level [158]. A collection of studies that report on or investigate the effect of antioxidant supplementation on semen variables and oxidative stress markers in men infected with SARS-CoV-2 are presented in Table 6.

## 6. Conclusions, Challenges, and Future Perspectives

There is enough evidence indicating an alteration of the male reproductive system after SARS-CoV-2 infection, though the duration of these defects is not well established. The high expression of ACE-2 receptor and TMPRSS-2 membrane protease on testicular cells seems to be a possible explanation for men’s health and male fertility’s greater susceptibility to COVID-19. Testicular (endocrine and exocrine) functions are among the organs beyond respiratory function affected by the COVID-19 disease. Evidence coming from semen analysis after the infection indicates that semen parameters are defective. Moreover, acute stage hypogonadism has also been observed, suggesting that the hormonal milieu is altered. Testicular tissue pathology from autopsies by COVID-19 deceased patients has also provided valuable data. Most studies report evidence of orchitis, vascular changes, basal membrane thickening, Leydig and Sertoli cell damage, and reduced spermatogenesis associated with SARS-CoV-2. However, studies investigating the presence of the virus in the ejaculate and testicular tissue report that this finding is relatively rare, leading us to conclude that other indirect mechanisms, except direct damage of the virus onto the testis, are involved. Oxidative stress is the backbone pathophysiologic mechanism associated with most indirect factors contributing to male infertility induced by COVID-19 diseases, such as inflammatory processes, cytokine storm, fever, antiviral drugs, and psychological stress. Oxidative and antioxidative markers of the seminal plasma reveal that SARS-CoV-2 infection impairs antioxidant defense machinery while ROS production is enhanced. The administration of antioxidant supplements has been put forth as a potential treatment of COVID-19-induced defects in male fertility and supplementary therapy to prevent disease progression in general. Antioxidants are commonly used as therapeutic agents to tackle OS-induced male infertility. However, their utility lacks critical evidence and must be further evaluated and interpreted cautiously. The excessive administration of antioxidants may disrupt the redox balance and induce reductive stress equally harmful to male reproduction as oxidative stress. Therefore, antioxidants should be cautiously administered before OS screening with frequent follow-up control of the redox state.

Nevertheless, there are a few studies indicating that antioxidants could be a promising treatment to support semen quality recovery and restore the redox balance after COVID-19, especially relevant for infected men currently seeking fertility and clinicians who are trying to improve the reproductive care for COVID-19 patients. There are also antioxidants; their use as a treatment for COVID-19 and their possible role in alleviating COVID-19-mediated damage in male fertility has been hypothesized. In that vein, future research should evaluate the possible combinations of antioxidant regimens, treatment duration, and exact dosage, as well as optimize administration timing. It is quite clear that we do not have solid evidence to adopt this therapeutic pathway to date broadly, and the efficacy of antioxidants such as Se, vitamin C, flavonoids, and NAC needs to be carefully addressed in future studies. Importantly, a careful andrological evaluation of men recovering from COVID-19 is also mandatory since evidence about the effects of COVID-19 on male fertility is somewhat concerning. Last but not least, there is no evidence about the harmful impact of COVID-19 vaccines on male fertility; counter wise, they have a protective role.

## Figures and Tables

**Figure 1 antioxidants-12-01483-f001:**
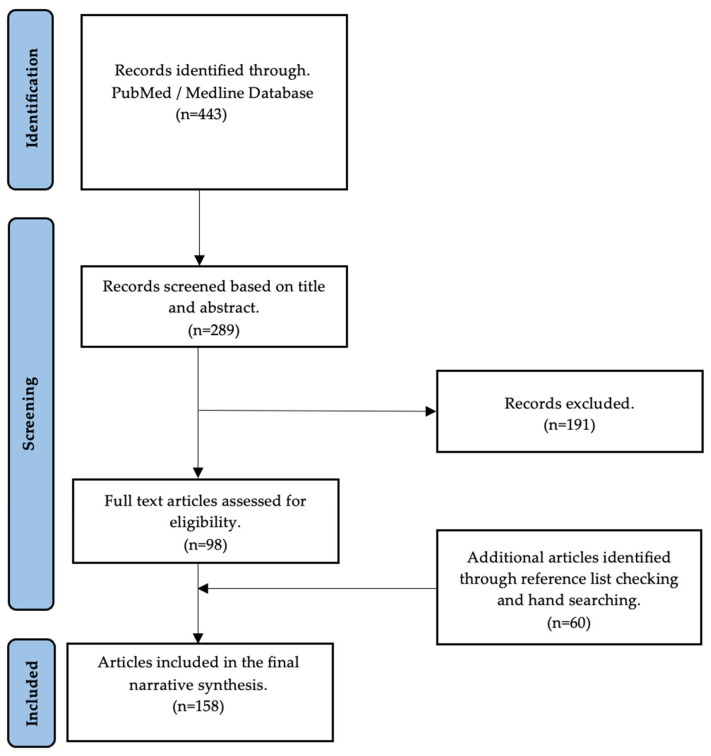
Flow diagram of studies included in the narrative review.

**Figure 2 antioxidants-12-01483-f002:**
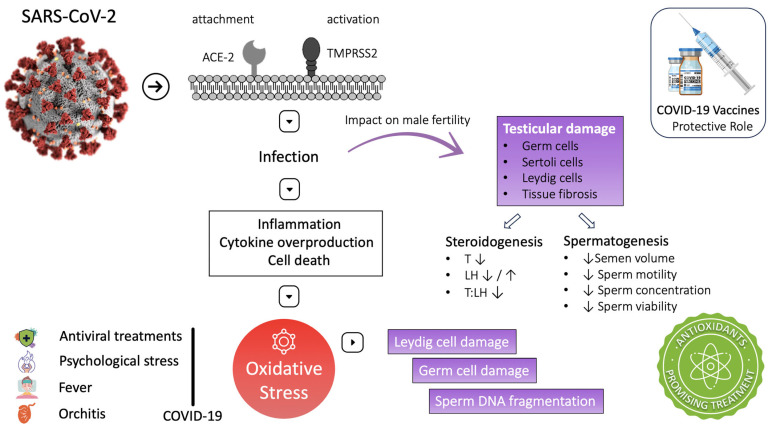
COVID-19 and male infertility: mechanisms and impact on fertility. ↑: increase ↓: decrease.

**Table 1 antioxidants-12-01483-t001:** Type of studies included in the narrative review.

TYPE	NUMBER	REFERENCES
REVIEWS	30	[3,4,8,12,19,20,21,22,23,24,25,26,27,28,29,30,31,32,33,34,35,36,37,38,39,40,41,42,43]
ORIGINAL RESEARCH	88	[1,10,13,14,15,16,17,18,44,45,46,47,48,49,50,51,52,53,54,55,56,57,58,59,60,61,62,63,64,65,66,67,68,69,70,71,72,73,74,75,76,77,78,79,80,81,82,83,84,85,86,87,88,89,90,91,92,93,94,95,96,97,98,99,100,101,102,103,104,105,106,107,108,109,110,111,112,113,114,115,116,117,118,119,120,121,122]
SYSTEMATIC REVIEW	3	[7,123,124]
SYSTEMATIC REVIEW AND METANALYSIS	6	[125,126,127,128,129,130]
LETTER TO THE EDITOR	9	[131,132,133,134,135,136,137,138,139]
CORRESPONDENCES	3	[11,140,141]
EDITORIAL	3	[6,142,143]
COMMUNICATION	3	[9,144,145]
COMMENTARY	4	[5,146,147,148]
OPINION	2	[149,150]
PERSPECTIVES	2	[151,152]
CASE REPORT	3	[153,154,155]
VIEWPOINT	1	[156]
**TOTAL**	**157**	

**Table 2 antioxidants-12-01483-t002:** Semen SARS-CoV-2 detection results in studies assessing COVID-19.

Author	Study Design	Sample Size, Stage	Specimen	Time since Diagnosis (Days)	Clinical Category	Results
Li et al. [10]	Cohort study	15 acute stage 23 recovery stage	Semen	6 to 16	NP	Positive 4 of 15 and 2 of 23
Machado et al. [55]	Cross-sectional study	15 active phase	Semen	2 to 8	2 asymptomatic, 13 mild symptoms	Positive 1 case
Delaroche et al. [56]	Prospective observational study	32 acute stage	Semen	0 to 1	NP	Positive 1 case in semen and seminal plasma not spermatozoa pellet
Saylam et al. [57]	Cross-sectional study	30 acute stage	Semen, urine	1	NP	Positive 4 semen samples, 7 urine samples
Kayaaslan et al. [58]	Cross-sectional study	16 acute stage	Semen	0 to 7	11 mild, 5 moderate disease	Negative
Rawlings et al. [59]	Cross-sectional study	6 acute or late phase	Semen	6 to 15	NP	Negative
Burke et al. [60]	Cohort study	18 acute and recovery phase	Semen	1 to 28	1 asymptomatic, 2 mild, 15 moderate	Negative
Guo et al. [61]	Cross-sectional study	23 acute and recovery stage	Semen	27 to 33	18 mild, 5 moderate	Negative
Holtman et al. [62]	Cohort study	18 recovery stage 2 acute stage	Semen	37 to 52	1 asymptomatic, 15 mild, 4 moderate	Negative
Ruan et al. [63]	Cohort study	74 recovery stage	Semen, urine, prostatic secretion	NP	11 mild, 31 moderate type, 32 severe pneumonia	Negative
Pan et al. [64]	Cross-sectional study	34 recovery stage	Semen	8 to 75	Mild, moderate symptoms	Negative
Ma et al. [14]	Cross-sectional	12 recovery stage	Semen	56 to 109	1 mild, 11 moderate type	Negative
Sharma et al. [65]	Prospective observational study	11 recovery stage	Semen	19 to 59	9 mild, 2 moderate	Negative
Best et al. [66]	Prospective observational study	30 recovery stage	Semen	11 to 64	1 asymptomatic, 29 symptomatic	Negative
Pavone et al. [140]	Cross-sectional	9 recovery phase	Semen	7 to 88	1 asymptomatic, 8 mild symptoms	Negative
Donders et al. [67]	Prospective cohort study	120 recovery stage	Semen	18 to 88	NA	Negative

NP: not provided. NA: not applicable.

**Table 3 antioxidants-12-01483-t003:** Studies investigating the effectiveness of antioxidant supplementation on male infertility.

Author	Study Characteristics	Vaccine Type	Study Group	Interval Time after Vaccination	Results
Abd et al. [79]	2 centre prospective observational study. Semen parameters pre and post vaccine.	mRNA (Pfizer)	60 healthy men with previous normal semen analysis	Second dose at least 90 days before the new semen analysis	No difference in semen parameters, except total and progressive motility. Clinically insignificant difference and within WHO normal ranges.
Diaz et al. [80]	Single center prospective study. Semen parameters pre- and post-vaccine (long follow-up).	mRNA (Pfizer, Moderna)	12 young healthy men	Follow up sample 3 and 9 months after 2nd dose	No changes in any semen parameters between baseline and follow-up samples.
Gonzalez et al. [81]	Single-center prospective study. Semen parameters pre- and post-vaccine (short follow-up).	mRNA (Pfizer, Moderna)	45 healthy men	Follow-up sample 70 to 86 days after 2nd dose	Semen volume, concentration, motility, and TMSC ↑ in the follow-up sample. Men with oligospermia did not show further decline.
Barda et al. [82]	Prospective observational cohort study. Semen parameters pre- and post-vaccine (short, long follow-up).	mRNA (Pfizer)	33 sperm donors	Multiple semen samples pre- and post-vaccine; at least one sperm sample 72 days after 2nd dose	TSC ↑, TMSC ↑, number of samples available for freezing ↑ and the presence of motile sperm after freezing ↑ post-vaccine.
Gat et al. [83]	Retrospective multicenter cohort study. Semen parameters pre- and post- vaccine (short, intermediate, and long evaluation).	mRNA (Pfizer)	37 semen donors	Multiple pre- and post-vaccine semen samples, at 3 time frames (15–45, 75–125, and over 145 days after 2nd dose)	Temporary decline of sperm concentration and TMC 3 months after vaccine, followed by recovery.
Karavani et al. [84]	Retrospective cohort study. Semen parameters pre- and post-vaccine (long follow-up).	mRNA (Pfizer)	58 men undergoing IVF (normal, abnormal semen analysis)	Follow-up sample 6 to 14 months after 1st dose	No difference in pre- and post-vaccine semen analysis, as well as subgroups with normal and abnormal semen parameters pre-vaccine.
Safrai et al. [85]	Retrospective cohort study. Semen parameters pre- and post-vaccine.	mRNA	72 men undergoing IVF (normal, abnormal semen analysis)	Follow-up sample 40 to 104 days after 1st dose	No changes pre- and post-vaccine among men with normal and abnormal semen analysis.
Olana et al. [86]	Single-center prospective study. Semen parameters, OS, inflammation, cell membrane activity pre- and post-vaccine.	mRNA (Pfizer)	47 subjects	Follow-up sample 3 months after 2nd dose	No difference in semen parameters, oxidative stress analysis, and IL-6 and electrolyte function pre- and post-vaccine.
Elhabak et al. [87]	Prospective cohort study. Semen characteristics pre- and post-vaccine (short follow-up).	Inactivated virus (Sinopharm), viral vector (AstraZeneca)	100 healthy men	Follow-up sample 70 days after 2nd dose	No differences in all semen characteristics pre- and post-vaccine.
Zhu et al. [88]	Retrospective cohort study. Semen parameters pre- and post-vaccine.	Inactivated virus	43 sperm donors	Follow-up sample 21 days after 1st dose and 60 days after 2nd dose	No changes in semen parameters pre- and post-vaccine.
Meitei et al. [89]	Retrospective observational study. Semen parameters pre- and post-vaccine.	Viral vector (Covishield)	53 subfertile men	Follow-up sample 83 days after 2nd dose	No variation in semen parameters pre- and post-vaccine, except morphology moderate ↓, clinically insignificant.
Orvieto et al. [90]	Observational study. Effect of vaccine on IVF cycle attempt.	mRNA	36 couples undergoing IVF	IVF cycles pre-vaccine and 7 to 85 days after 2nd dose	No difference in fertilization rate, pregnancy rate, and number of oocytes retrieved pre- and post- vaccination.
Reschini et al. [91]	Multicenter retrospective study. Semen parameters and fertilization rate pre- and post-vaccine.	mRNA (Pfizer, Moderna), viral vector (AstraZeneca, Johnson & Johnson)	106 men undergoing ART	ART cycles pre-vaccine and 28 to 100 days after 2nd dose	No difference in fertilization rate and sperm analysis.
Xia et al. [92]	Cohort study. Impact of vaccine on IVF cycles.	Inactivated virus (Sinovac, Sinopharm)	Vaccinated men (*n* = 105); unvaccinated men (*n* = 155)	NP	No difference in semen parameters, IVF laboratory outcomes, and pregnancy rates between 2 groups.
Wang et al. [93]	Single-center prospective cohort study. Impact of vaccine on IVF outcomes.	Inactivated virus (Sinovac)	Vaccinated men (*n* = 275); unvaccinated men (*n* = 944)	NP	No difference in laboratory and clinical outcomes during ART between 2 groups.

NP: not provided. ↑: increase ↓: decrease.

**Table 4 antioxidants-12-01483-t004:** Studies investigating the impact of COVID-19 on seminal biomarkers and testicular health.

Author	Study Characteristics	Sample Size—Disease Severity	Specimen	Time	Results
Hajizadeh Maleki et al. [94]	Prospective cohort study. Impact on multiple seminal biomarkers.	84 recovering fertile men—different severity, 105 healthy controls	Semen	Hospital discharge, every 10 days until 2 months	ROS levels ↑, pro- and anti-inflammatory cytokines ↑, apoptotic variables ↑, semen ACE-2 enzymatic activity ↑, SOD activity ↓ ↔ defective semen quality, disease severity
Falahieh et al. [95]	Prospective cohort study. Impact on semen oxidative status and parameters.	20 recovering fertile men—moderate disease	Semen	14 and 120 days after COVID-19 diagnosis	14 days: sperm morphology, total and progressive motility ↓, peroxidative-positive leucocytes ↑, DFI ↑, ROS ↑, MDA ↑, TAC ↓ 120 days: all parameters improved
Gharagozloo et al. [154]	Case report. Longitudinal analysis of semen quality.	Recovering man—moderate disease	Semen	Pre- and post-COVID-19	Spermatogenesis disruption, oxidative DNA damage levels ↑
Shcherbitskaia et al. [96]	Prospective cohort study. Oxidative stress markers and sperm DNA fragmentation.	17 recovering men, 22 controls	Semen	5 months after infection	COVID-19 not always correlated with DNA fragmentation levels; COVID-19 patients with abnormal TUNNEL rate → NT ↓, TAC ↓, Zn ↓, 8-OHdG ↑, round cell numbers ↑
Moghimi et al. [97]	Incidence of apoptosis within the testes of patients succumbed from COVID-19.	6 COVID-19 deceased men, 6 controls	Testicular tissue	20 to 32 days disease duration	Seminiferous tubules, interstitial tissue damage, testicular cells ↓, ↑ ROS, ↓ GSH Ace-2, caspase-3, BAX ↑–BCL-2 expression ↓, apoptotic cells % ↑
Flaifel et al. [133]	Morphologic features in testes obtained from patients with COVID-19.	10 COVID-19 deceased men	Testicular, epididymis tissue	7 to 27 days disease duration	Morphologic alterations attributable to OS (chromatin condensation, nuclear fragmentation, and acidophilic cytoplasm); sloughing of spermatocytes into the tubular lumen and accumulation in the epididymis, spermatids elongation, damage of Sertoli cells, multifocal microthrombi, ↑ platelets in testicular vessels, ↑ mononuclear inflammatory infiltrates in the interstitial and other evidence related to OS

↑: increase ↓: decrease ↔: no change, →: progression.

**Table 5 antioxidants-12-01483-t005:** Overview of studies assessing the effectiveness of antioxidant supplementation on male infertility.

Author	Study Type, Objectives	Antioxidants	Study Groups	Results
Verma et al. [100]	Sperm motility, viability, and lipid peroxidation assessed in Ringer–Tyrode supplemented with different concentrations of ascorbic acid.	Ascorbic acid	Samples with motility higher than 60% and sperm count over 20 million/mL were used.	Concentrations below 1000 μΜ of ascorbic acid increased sperm motility and decreased lipid peroxidation. However, concentrations above 1000 μM caused the complete reverse effect.
Menezo et al. [101]	DNA fragmentation index and the degree of sperm decondensation were assessed before and after antioxidant supplementation at specific doses.	Vitamins C and E, β-carotene, zinc, and selenium	58 patients who had at least 2 IVF or ICSI failure attempts were included.	DNA fragmentation index decreased and DNA decondensation increased after 90 days of treatment.
Bleau et al. [102]	Association between selenium concentrations in semen and semen quality.	Selenium	125 men from couples consulted for infertility.	Se levels <35 ng/mL were corelated with male infertility. Se levels between 40 and 70 ng/mL were optimal. Se levels >80 ng/mL were associated with a high abortion rate, decreased motility, and asthenospermia.
Balercia et al. [103]	Placebo-controlled, double-blind randomized trial. Effectiveness of antioxidants in improving semen quality in men with idiopathic infertility.	Coenzyme Q10 (CoQ10)	55 patients with idiopathic infertility.	CoQ10 and ubiquinol were increased in semen, and sperm kinetic features were improved after treatment. In the treated group, six pregnancies were observed, twice as many as those observed in the control group.
Balercia et al. [104]	Placebo-controlled, double-blind, randomized trial. Effectiveness of antioxidants in improving semen kinetic parameters and the total oxyradical scavenging capacity in semen.	L-carnitine (LC) or L acetylcarnitine (LAC) or combined LC and LAC	59 patients with infertility.	Sperm cell motility improvement was detected in patients administered with LAC single treatment or in combination with LC. TOSC assay improved hydroxyl and peroxyl radicals for treated patients. Nine pregnancies were achieved during the therapy in the treated group.
Busetto et al. [105]	Monocentric, randomized, double-blind, placebo-controlled trial. Effect of antioxidants on sperm quality.	LC, LAC, fructose, fumarate, vitamin C, vitamin B12 citric acid, zinc, selenium, CoQ10	94 patients with oligo- and/or astheno- and/or teratozoospermia with or without varicocele.	Sperm concentration, total sperm count, and total-progressive motility were increased in the supplemented group. Ten pregnancies were recorded during the follow-up period in the treated group.
Tremellen et al. [106]	Prospective randomised double-blind placebo-controlled trial. Effect of antioxidant supplementation on embryo quality and pregnancy outcome during IVF-ICSI.	Vitamins C and E, selenium, garlic lycopene, zinc, and folic acid	60 couples with severe male factor infertility.	Improvement in viable pregnancy rate was found in the antioxidant-administered group. No improvement was detected in the two groups in oocyte fertilization and embryo quality rates.
Salvio et al. [123]	Systematic review. Semen quality assessed by conventional, advanced methods, and pregnancy rates to determine CoQ10 therapy usefulness in infertile men. 24 studies included.	CoQ10 alone or in combination with other antioxidants	-	Improvement in sperm quality, especially sperm motility, was noted after CoQ10 administration.
Tsampoukas et al. [124]	Systematic review. The role of L-carnitine in the treatment of varicocele. Four studies included.	L-carnitine administration alone or in duet	-	Semen parameters of men with varicocele-related infertility were improved after antioxidant supplementation.
Greco et al. [107]	Randomized, placebo-controlled, double-blind study. Reduction of the incidence of sperm DNA Fragmentation after antioxidant treatment.	Vitamins C and E	64 men with unexplained infertility and elevated percentage of DNA fragmented spermatozoa in the semen.	No differences in basic sperm parameters were found. DNA-fragmented spermatozoa reduced in the antioxidant treatment group.
Rolf et al. [108]	Single-centre randomized, placebo-controlled, double-blind study. Effect of antioxidants intake on semen parameters of infertile men.	Vitamins C and E	31 men with asthenozoospermia or moderate oligoasthenozoospermia.	No changes in semen parameters were observed during treatment, and no pregnancies were initiated during the treatment period.
Pyrgidis et al. [126]	Systematic review and meta-analysis. The effect of antioxidant intake on operated or non-operated varicocele-associated infertility. Fourteen studies included.	Various antioxidants	-	Antioxidant did not seem to improve pregnancy rates, DNA fragmentation, and semen parameters in patients with varicocele-related infertility.
Showell et al. [127]	Systematic review and meta-analysis. Effectiveness and safety of antioxidants intake for subfertile male partners in couples seeking fertility assistance. Thirty-four RCTs included.	Various antioxidants	-	Antioxidant therapy positively affected the number of pregnancies and live births in sub-fertile couples undergoing ART treatment.
Smits et al. [128]	Meta-analysis. Effectiveness and safety of antioxidants in subfertile men. Sixty-one RCTs included.	18 different oral antioxidants	-	Antioxidants may lead to increased live birth and clinical pregnancy rates.
Humaidan et al. [109]	Prospective study. The combined effect of lifestyle changes and antioxidant therapy on sperm DNA fragmentation and seminal OS in IVF patients.	Coenzyme Q10, omega-3, and oligo-elements	93 infertile males with a history of failed IVF/ICSI. 10 healthy male volunteers as controls.	No differences in semen quality and sORP were observed.
Steiner et al. [157]	Multi-center, double blind, randomized, placebo-controlled trial with an internal pilot study. Effect of antioxidants intake on semen parameters, DNA fragmentation and live birth.	Vitamins C and E, selenium, L-carnitine, zinc, folic acid, and lycopene	174 infertile men. Attempts to conceive naturally the first 3 months and with clomiphene citrate with intrauterine insemination in months 4 to 6.	No difference in the cumulative live birth rate was observed. The interval pilot study that assessed the semen parameter and DFI changes detected no improvement.
Knudtson et al. [110]	Secondary analysis of randomized clinical trial. Relationship of plasma antioxidant levels to semen parameters.	Vitamin E, zinc, or selenium	Men attending fertility centers.	No association between selenium, zinc, or vitamin E levels and semen parameters or DNA fragmentation.
Khaw et al. [111]	Systematic review and meta-analysis. Safety and efficacy of carnitine supplementation for idiopathic male infertility. Seven studies included.	L-carnitine and L-acetylcarnitine	-	Carnitines improved total sperm motility, progressive sperm motility, and sperm morphology. No effect on clinical pregnancy rate in the five studies that included that outcome.
Zhou et al. [111]	Systematic review and meta-analysis. Effect of antioxidants on sperm parameters and serum hormones in idiopathic infertile men. Three RCTs included.	N-acetyl-cysteine	-	Improvement in sperm concentration, ejaculate volume, sperm motility, and normal morphology. No significant influence in serum hormones.
Szymanski et al. [112]	Retrospective study. The use of antioxidants to improve qualitative and quantitative deficiencies in the male gametes.	LC, LAC vitamin C, coenzyme Q10, zinc, folic acid, selenium, and vitamin B12	78 men with idiopathic infertility.	Improvement in semen parameters, except the percentage of sperm of abnormal morphology and semen volume after treatment.
Nazari et al. [113]	Prospective interventional study. Efficacy of antioxidant supplementation on semen parameters.	Antioxidant supplements containing L-carnitine	59 infertile male patients with idiopathic oligoastenoteratozoospermia.	Improvement in the sperm Concentration and sperm morphology. Sperm motility was not altered after treatment.
Wei et al. [130]	Metanalysis. Efficacy of antioxidant supplementation in men with idiopathic asthenozoospermia. Seven RCTs were included.	L-carnitine, L-acetylcarnitine, and N-acetyl-cysteine	-	LC/LAC and NAC improved sperm motility and normal morphology. NAC positively affected sperm concentration and semen volume.
Alahmar et al. [114]	Prospective controlled clinical study. Determination of the biochemical and clinical predictors of pregnancy and time to pregnancy in infertile patients with idiopathic oligoasthenospermia before and after CoQ10 supplementation.	Coenzyme Q10	178 male patients with idiopathic oligoasthenospermia and 84 fertile men (controls).	CoQ10 intake increased CoQ10 levels in seminal plasma and ameliorated semen parameters, SDF, and antioxidant capacity with a pregnancy rate of 24.2%. CoQ10 levels, semen parameters, ROS, GPx, and male age could be used as diagnostic biomarkers for male fertility and predictors for time to pregnancy and pregnancy outcome.
Alahmar et al. [115]	Prospective controlled study. Effect of CoQ10 on sperm DNA damage and OS markers in infertile men with idiopathic oligoastenoteratozoospermia.	Coenzyme Q10	Fifty patients with idiopathic oligoastenoteratozoospermia and 50 fertile men (controls).	Improved sperm quality and seminal antioxidant status and reduced total ROS and SDF levels after treatment compared to pre-treatment values.
Maghsoumi-Norouzabad et al. [116]	Randomized, triple blind, placebo-controlled clinical trial. Effect of vitamin D3 on semen parameters and endocrine markers in infertile patients with asthenozoospermia.	Vitamin D3	86 infertile men with asthenozoospermia and serum 25 hydroxy vitamin D3 <30 ng/mL in the infertility clinic.	Increased serum 25 hydroxy vitamin D3, parathyroid hormone, phosphorus, seminal and serum calcium, T/LH ratio, and total and progressive sperm motility in infertile men with asthenozoospermia.

**Table 6 antioxidants-12-01483-t006:** Studies reporting or investigating the effect of antioxidant supplementation on semen variables and oxidative stress markers in men infected with SARS-CoV-2.

**Author,** **Year**	**Study Type,** **Objective**	**Study Group**	**Treatment Regimen**	**Treatment Results**
Rafiee et al. 2021 [117]	Interventional study. Εffect of NAC on abnormal semen parameters in men infected by COVID-19 the last 2 months.	Two-hundred men referred to infertility clinics due to female factor infertility, diseased by COVID-19. Divided into 2 groups: supplemented group (*n* = 100), and the control group (*n* = 100).	N-acetylcysteine	Sperm concentration ↑ *, sperm total motility ↑ *, sperm morphology ↑ *; semen parameters were comparable to pre-COVID-19 values in the supplemented group. Not significant in the control group.
Kurashova et al. 2022 [118]	Original paper. Efficacy of an antioxidant regimen on men with pathozoospermia after COVID-19 infection.	Twenty-five men with pathozoospermia pre-existing COVID-19.	Astaxanthin, Omega-3 fatty acids	Sperm motility ↑ *, number of leucocytes in semen ↓ *, TBARS ↓ *, and AOA levels ↑ * in the blood after treatment.
Mannur et al. 2021 [155]	Case report.	Normozoospermic man undergoing infertility treatments due to female fertility factor developed oligo-astheno-teratozospermia with severe sperm DNA damage 1 month after recovery from mild COVID-19.	Coenzyme Q10, lycopene, docosahexaenoic acid, folic acid, selenium, and zinc	Sperm count ↑, sperm motility ↑; however, morphology and sperm DNA damage remained defected after treatment.
Aschauer et al. 2023 [119]	Prospective comparative pilot study. Efficacy of micronutrient supplementation to improve seminal OS and semen variables after symptomatic SARS-CoV-2 infection.	Forty men without subfertility history recently recovered from symptomatic COVID-19. Suplemented group (*n* = 30), and the control group (*n* = 10).	L-carnitine, L-arginine, vitamin E, glutathione, coenzyme Q10, folic acid, selenium, and zinc	Sperm total and progressive motility ↑ *, sperm vitality ↑ *, and the number of patients with normal semen analysis values ↑ * in the study group. No changes in the control group. sORP levels ↓ in both groups, with the decrease more evident in the study group.

***** statistically significant changes.

## Data Availability

Not applicable.

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
