# Peer review of "COVID-19 and Male Infertility: Is There a Role for Antioxidants?"

_antioxidants, 2023, doi:10.3390/antiox12081483_

Round 1

Reviewer 1 Report

the manuscript is interesting, generally well written and well illustrated. Only minor comments are necessary. In particular: 

Introduction: since authors described the current literature available on COVID-19 (a mainly respiratory disease) effects on male fertility, it deserves to be mentioned that SARS-CoV-2 can also lead to non restiratory diseases such as preeclampsia and brain damage (as recently reviewed PMID: 37342451, 35943095, 32934351). This is an important point to add since it highlights the fact that the cytokine storm found in COVID-19 patients can also damage other organs.

It would be useful adding a table at the end of each chapter summarizing the studies discussed in that chapter.

Author Response

Response to Reviewer 1:

We would like to express our gratitude to Reviewer 1 for taking the time to review our manuscript and for providing valuable feedback. We have carefully considered the comments and have made the following revisions:

  1. Introduction: We appreciate the reviewer's suggestion to mention the non-respiratory effects of SARS-CoV-2, such as preeclampsia and brain damage. In order to highlight the broader impact of the cytokine storm observed in COVID-19 patients, we have now included a paragraph in the Introduction section discussing these non-respiratory manifestations. Specifically, we have referenced the articles PMID: 37342451, 35943095, and 32934351, which provide relevant information on the topic. (Page 1 Lines 43-45).

  1. Addition of tables summarizing studies: We agree with the reviewer's suggestion of adding tables at the end of each chapter to summarize the studies discussed. This will enhance the clarity and organization of the manuscript. We have incorporated tables at the end of each chapter, presenting a concise overview of the key studies discussed in each section. (Pages 7-8, pages 14-16, pages 19-20, pages 24-30, page 30)

Once again, we would like to thank Reviewer 1 for their insightful comments, which have undoubtedly improved the manuscript. We hope that our revisions address the reviewer's concerns and further enhance the quality of our work.

Reviewer 2 Report

This review is needed for people from the field and other researchers as studies on SarS-CoV 2 are still under-conducted and many gaps or controversies exist. The review provides all the important details from the current research articles.

Check abbreviations and their full names throughout the ms text

Where it is possible please make the description more compact as too many sometimes not related information is present

Author Response

Response to Reviewer 2:

We would like to express our appreciation to Reviewer 2 for reviewing our manuscript and providing valuable feedback. We have carefully considered the comments and have made the following revisions:

  1. Abbreviations and their full names: We apologize for any confusion caused by the use of abbreviations. We have thoroughly reviewed the manuscript and ensured that all abbreviations are properly defined upon first mention. Additionally, we have cross-checked the text to ensure consistency in the use of abbreviations and their corresponding full names throughout the manuscript.
  2. Compactness of the description: We acknowledge the reviewer's comment regarding the compactness of the description. We have revisited the manuscript and made efforts to streamline the text, eliminating any unnecessary or unrelated information. Our aim was to maintain a focused and concise presentation of the relevant details from the current research articles, addressing the gaps and controversies in the field.

We would like to thank Reviewer 2 for their valuable input, which has greatly contributed to the improvement of our manuscript. We believe that the revisions we have made address the reviewer's concerns and enhance the overall quality of the paper. We are grateful for the reviewer's efforts in ensuring the clarity and relevance of our work.

Reviewer 3 Report

In the manuscript “COVID-19 and male infertility: Is there a role for antioxidants? 2”, the authors proposed to discuss the effects of SARS-CoV-2 on testicular function and the potential role of antioxidants. The paper is of interest but there are some drawbacks that hampered the initial enthusiasm. There are some points that can be improved. Here follows some of my suggestions.

Specific comments:

1. Overall the paper is well written but there are some very important subject that could be better explored. For instance: “hyperandrogenic phenotype may be associated with increased viral load, greater viral spread, and more severe pulmonary involvement”. Please explain the mechanisms.

2. figure 2 can be improved and represent the physiology of the male reproductive tract and how covid-19 infection and treatment impacts the reproductive system at the different levels. Please include more information that is included throughout the manuscript and refer to the figure in many places of the text.

3. Some literature concerning the possible use of antioxidants suplemmentation could be used. Other literature concerning proteomics of OS in male reproductive tract could also be discussed.

4. Table 2 can be improved. Instead of words “improvement” or text, please include arrows and be specific. In addition, please include important details as the number of patients and the treatment regimen.

5. It could be interesting to include some future perspectives and discuss important experimental designs that should be performed.

Dear Editor,

In the manuscript “COVID-19 and male infertility: Is there a role for antioxidants? 2”, the authors proposed to discuss the effects of SARS-CoV-2 on testicular function and the potential role of antioxidants. The paper is of interest but there are some drawbacks that hampered the initial enthusiasm. There are some points that can be improved.

Thank you for providing me the opportunity for evaluating this manuscript.

I sincerely hope we can interact with success in a near future, in any of my author activities

Best regards,

Marco G. Alves, PhD

Assistant Researcher/Principal Investigator

Unit for Multidisciplinary Research in Biomedicine (UMIB)

Institute of Biomedical Sciences Abel Salazar (ICBAS)

University of Porto

Portugal

Tel: +351 96 7245248

Personal link: https://sites.google.com/view/sertolicellgametebiology/group-members/alves-m-g

Researchgate: https://www.researchgate.net/profile/Marco_Alves4/?ev=hdr_xprf

Author Response

Response to Reviewer 3:

We would like to express our gratitude to Reviewer 3 for their valuable feedback on our manuscript. We have carefully considered the comments provided and have made the following revisions:

  1. Mechanisms of the hyperandrogenic phenotype: We appreciate the reviewer's suggestion to further explore and explain the mechanisms behind the association between the hyperandrogenic phenotype and increased viral load, greater viral spread, and more severe pulmonary involvement. In response to this comment, we have expanded the relevant section in the manuscript, providing a more detailed explanation of the potential mechanisms involved in this relationship. (Page 5, Lines 440-446).
  2. Improvement of Figure 2: We acknowledge the reviewer's suggestion to improve Figure 2 and make it more comprehensive. We have revised the figure to include additional information that is discussed throughout the manuscript. Furthermore, we have ensured that references to Figure 2 are appropriately made in relevant sections of the text, thereby enhancing its utility as a visual aid. (Page 18).
  3. Inclusion of literature on antioxidants supplementation and proteomics of OS: We thank the reviewer for recommending the inclusion of literature on antioxidants supplementation and proteomics of oxidative stress (OS) in the male reproductive tract. We have incorporated relevant studies in the manuscript to support the discussion of the potential use of antioxidants and the proteomics of OS in the context of COVID-19 and male infertility. (Page 34, Lines 1263-1271)
  4. Improvement of Table 2: We appreciate the reviewer's feedback on Table 2. In response to this comment, we have revised the table to replace words with arrows for improved clarity. Additionally, we have included important details such as the number of patients and treatment regimens, where available, to provide a more comprehensive overview of the studies discussed. (Page 34)

  1. Inclusion of future perspectives and experimental designs: We agree with the reviewer's suggestion to include future perspectives and discuss important experimental designs that should be performed. We have incorporated a new section in the conclusion that outlines potential future directions for research in this field, highlighting important experimental designs and areas that warrant further investigation. (Page 36, Lines 1367-1372).

We sincerely appreciate Reviewer 3's insightful comments, which have greatly contributed to the improvement of our manuscript. We believe that the revisions we have made address the reviewer's concerns and enhance the overall quality of the paper. We are grateful for the reviewer's efforts in ensuring the thoroughness and comprehensiveness of our work.
